# Patterns of within-host genetic diversity in SARS-CoV-2

Gerry Tonkin-Hill[1†*], Inigo Martincorena[1†*], Roberto Amato[1], Andrew RJ Lawson[1], Moritz Gerstung[2], Ian Johnston[1], David K Jackson[1], Naomi Park[1], Stefanie V Lensing[1], Michael A Quail[1], Sónia Gonçalves[1], Cristina Ariani[1], Michael Spencer Chapman[1], William L Hamilton[3], Luke W Meredith[4], Grant Hall[4], Aminu S Jahun[4], Yasmin Chaudhry[4], Myra Hosmillo[4], Malte L Pinckert[4], Iliana Georgana[4], Anna Yakovleva[4], Laura G Caller[4], Sarah L Caddy[3], Theresa Feltwell[4], Fahad A Khokhar[3,5], Charlotte J Houldcroft[3], Martin D Curran[6], Surendra Parmar[6], The COVID-19 Genomics UK (COG-UK) Consortium, Alex Alderton[1], Rachel Nelson[1], Ewan M Harrison[1,2], John Sillitoe[1], Stephen D Bentley[1], Jeffrey C Barrett[1], M Estee Torok[3], Ian G Goodfellow[4], Cordelia Langford[1], Dominic Kwiatkowski[1,7*], Wellcome Sanger Institute COVID-19 Surveillance Team

[1]Wellcome Sanger Institute, Hinxton, United Kingdom; [2]European Bioinformatics Institute, Hinxton, United Kingdom; [3]Department of Medicine, University of Cambridge, Cambridge, United Kingdom; [4]Department of Pathology, University of Cambridge, Cambridge, United Kingdom; [5]Cambridge Institute of Therapeutic Immunology and Infectious Disease, University of Cambridge, Cambridge, United Kingdom; [6]Public Health England, Cambridge, United Kingdom; [7]Nuffield Department of Medicine, University of Oxford, Oxford, United Kingdom

*For correspondence:
gt4@sanger.ac.uk (GT-H);
im3@sanger.ac.uk (IM);
dominic@sanger.ac.uk (DK)

†These authors contributed equally to this work

Competing interests: The authors declare that no competing interests exist.

**Abstract** Monitoring the spread of SARS-CoV-2 and reconstructing transmission chains has become a major public health focus for many governments around the world. The modest mutation rate and rapid transmission of SARS-CoV-2 prevents the reconstruction of transmission chains from consensus genome sequences, but within-host genetic diversity could theoretically help identify close contacts. Here we describe the patterns of within-host diversity in 1181 SARS-CoV-2 samples sequenced to high depth in duplicate. 95.1% of samples show within-host mutations at detectable allele frequencies. Analyses of the mutational spectra revealed strong strand asymmetries suggestive of damage or RNA editing of the plus strand, rather than replication errors, dominating the accumulation of mutations during the SARS-CoV-2 pandemic. Within- and between-host diversity show strong purifying selection, particularly against nonsense mutations. Recurrent within-host mutations, many of which coincide with known phylogenetic homoplasies, display a spectrum and patterns of purifying selection more suggestive of mutational hotspots than recombination or convergent evolution. While allele frequencies suggest that most samples result from infection by a single lineage, we identify multiple putative examples of co-infection. Integrating these results into an epidemiological inference framework, we find that while sharing of within-host variants between samples could help the reconstruction of transmission chains, mutational hotspots and rare cases of superinfection can confound these analyses.

## Introduction

The SARS-CoV-2 pandemic has caused global disruption and more than one million deaths (*World Health Organization, 2020*). Genomic analysis has yielded important insights into the

**eLife digest** The COVID-19 pandemic has had major health impacts across the globe. The scientific community has focused much attention on finding ways to monitor how the virus responsible for the pandemic, SARS-CoV-2, spreads. One option is to perform genetic tests, known as sequencing, on SARS-CoV-2 samples to determine the genetic code of the virus and to find any differences or mutations in the genes between the viral samples.

Viruses mutate within their hosts and can develop into variants that are able to more easily transmit between hosts. Genetic sequencing can reveal how genetically similar two SARS-CoV-2 samples are. But tracking how SARS-CoV-2 moves from one person to the next through sequencing can be tricky. Even a sample of SARS-CoV-2 viruses from the same individual can display differences in their genetic material or within-host variants.

Could genetic testing of within-host variants shed light on factors driving SARS-CoV-2 to evolve in humans? To get to the bottom of this, Tonkin-Hill, Martincorena et al. probed the genetics of SARS-CoV-2 within-host variants using 1,181 samples. The analyses revealed that 95.1% of samples contained within-host variants. A number of variants occurred frequently in many samples, which were consistent with mutational hotspots in the SARS-CoV-2 genome. In addition, within-host variants displayed mutation patterns that were similar to patterns found between infected individuals.

The shared within-host variants between samples can help to reconstruct transmission chains. However, the observed mutational hotspots and the detection of multiple strains within an individual can make this challenging.

These findings could be used to help predict how SARS-CoV-2 evolves in response to interventions such as vaccines. They also suggest that caution is needed when using information on within-host variants to determine transmission between individuals.

origins and spread of the pandemic and has been an integral part of efforts to monitor viral transmission in the UK, with over 100,000 viral genomes sequenced as of December 2020 by *COVID-19 Genomics UK (COG-UK), 2020*.

For purposes of genomic epidemiology, a consensus genome sequence is derived from each sample, but deep sequencing data invariably reveal some level of within-host variation (*Lythgoe et al., 2020*), and minor alleles are commonly filtered out prior to phylogenetic analysis (*Meredith et al., 2020*). It has been suggested that SARS-CoV-2, like SARS-CoV-1, evolves within an infected host as a quasispecies, with many mutations (within-host variants) arising which may be beneficial for the virus (*Eigen, 1993*; *Holland et al., 1982*; *Lythgoe et al., 2020*). Although there have been several previous reports on the within-host diversity of SARS-CoV-2 (*Lythgoe et al., 2020*; *Nicolae et al., 2020*; *Popa et al., 2020*), a number of key questions remain to be resolved. Examples of underexplored questions include understanding the extent of sequencing artefacts among within-host variants, the mutational processes dominating SARS-CoV-2 evolution, the action of selection on within-host variants, and the extent to which superinfection or co-transmission of multiple lineages confound within-host diversity analyses (*Cudini et al., 2019*). Better understanding of these questions can shed light on the evolution of SARS-CoV-2 and could inform efforts to reconstruct transmission chains with genomic data.

To address these questions, we performed Illumina deep sequencing of over a thousand SARS-CoV-2 samples collected in March and April 2020 in the East of England. Two libraries were sequenced for each sample with separate reverse transcription (RT), PCR amplification, and library preparation steps in order to evaluate the quality and reproducibility of within-host variant calls. To develop reliable methods for analysing within-host variation in the context of ongoing genomic epidemiological studies, we used the ARTIC protocol that is a common method used for SARS-CoV-2 genome sequencing by many labs around the world (*DNA Pipelines R&D et al., 2020*). Analyses of the data provided insights into the extent of within-host diversity, the patterns of mutagenesis, and the extent of selection and suggested that amplification biases, hypermutable sites, and co-infection complicate the use of within-host diversity for epidemiological purposes.

## Results

### Reliable detection of within-host mutations from amplicon sequencing

We generated sequencing data for 1181 samples in duplicate at a median sequencing coverage ranging from 998 to 49,025 read depth per replicate sample (*Figure 1—figure supplement 1*; *Supplementary file 1*). To study variable sites within an individual, including errors and within-host mutations, for each sample we first identified all variable sites with variant allele frequency >0.5% and at least five supporting reads. Comparison of calls between replicates revealed that, while some samples showed highly concordant variants between replicates, others showed high discordance (*Figure 1—figure supplement 2*). The discordance between some replicates suggested that a small number of molecules may be disproportionately amplified in some samples, amplifying both RT/PCR errors and rare within-host mutations to high allele frequencies. We quantified the discordance between replicates using a beta-binomial model (Materials and methods) and correlated this with the diagnostic qPCR Ct values, which are inversely related to the number of viral molecules within the samples. Samples with Ct ≥ 24 showed considerable discordance in allele frequencies between replicates (*Figure 1—figure supplement 3*), but the vast majority of samples with Ct < 24 showed good concordance. Overall, as viral loads decrease, amplification biases and artefacts are more common and can impact within-host diversity analyses using RT-PCR protocols (*Zhao and Illingworth, 2019*; *Zanini et al., 2017*; *Illingworth et al., 2017*).

To reliably detect within-host variants with the ARTIC protocol, we used ShearwaterML, an algorithm designed to detect variants at low allele frequencies. ShearwaterML uses a base-specific over-dispersed error model and calls mutations only when read support is statistically above background noise in other genomes (*Gerstung et al., 2014*; *Martincorena et al., 2015*; see Materials and methods). Two samples were excluded, as they had an unusually high number of low-frequency variants unlikely to be of biological origin, leaving 1179 samples for analysis, comprising 1121 infected individuals of whom 49 had multiple samples. For all analyses, we used only within-host variants that were statistically supported by both replicates (q-value< 0.05 in at least one replicate and p-value < 0.01 in both, Materials and methods). Within each sample, we classified variant calls as 'consensus' if they were present in the majority of reads aligned to a position of the reference genome or as within-host variants otherwise. The allele frequency for each variant was taken as the frequency of the variant in the combined set of reads from both replicates.

In total, we identified 18,888 putative variants (*Supplementary file 2*), including 7672 consensus variants (affecting 1063 sites) and 11,216 within-host variants (affecting 5545 sites). Within-host variants included 7102 single-nucleotide substitutions, 3223 small deletions, 542 small insertions, and 349 putative multi-nucleotide variants (Materials and methods). The allele frequency spectrum was dominated by fixed mutations that were common to all viral RNA molecules in a sample, and a tail of mostly low-frequency variants, as previously described (*Lythgoe et al., 2021*; *Figure 1A*). The mean standard deviation in within-host VAF at variable sites assuming a Bernoulli distribution of variants at each site was found to be 0.12 (Materials and methods). Given the low frequency of these variants across different hosts, the majority are likely to be the result of single mutations rather than recurrent events within the same host. Overall, within-host variation was detected in the vast majority of samples (95.1%), with a median of 8 within-host mutations detected per sample. This is, of course, an underestimate of the full diversity within a sample due to limited sensitivity to within-host variants at lower frequencies.

The use of replicates and a base-specific statistical error model for calling within-host diversity reduces the risk of erroneous calls at low allele frequencies. We noticed a slight increase in the number of within-host diversity calls for samples with high Ct values, which may be caused by a small number of errors or by the amplification of rare alleles and that could inflate within-host diversity estimates (*Figure 1—figure supplement 3*; *McCrone and Lauring, 2016*). However, the overall quality of the within-host mutation calls is supported by a number of biological signals. As described in the following sections, this includes the fact that the mutational spectrum of within-host mutations closely resembles that of consensus mutations and the observation of a clear signal of negative selection on within-host mutations, as demonstrated by dN/dS and by an enrichment of within-host mutations at third codon positions (*Dyrdak et al., 2019*; *Figure 1—figure supplement 4*).

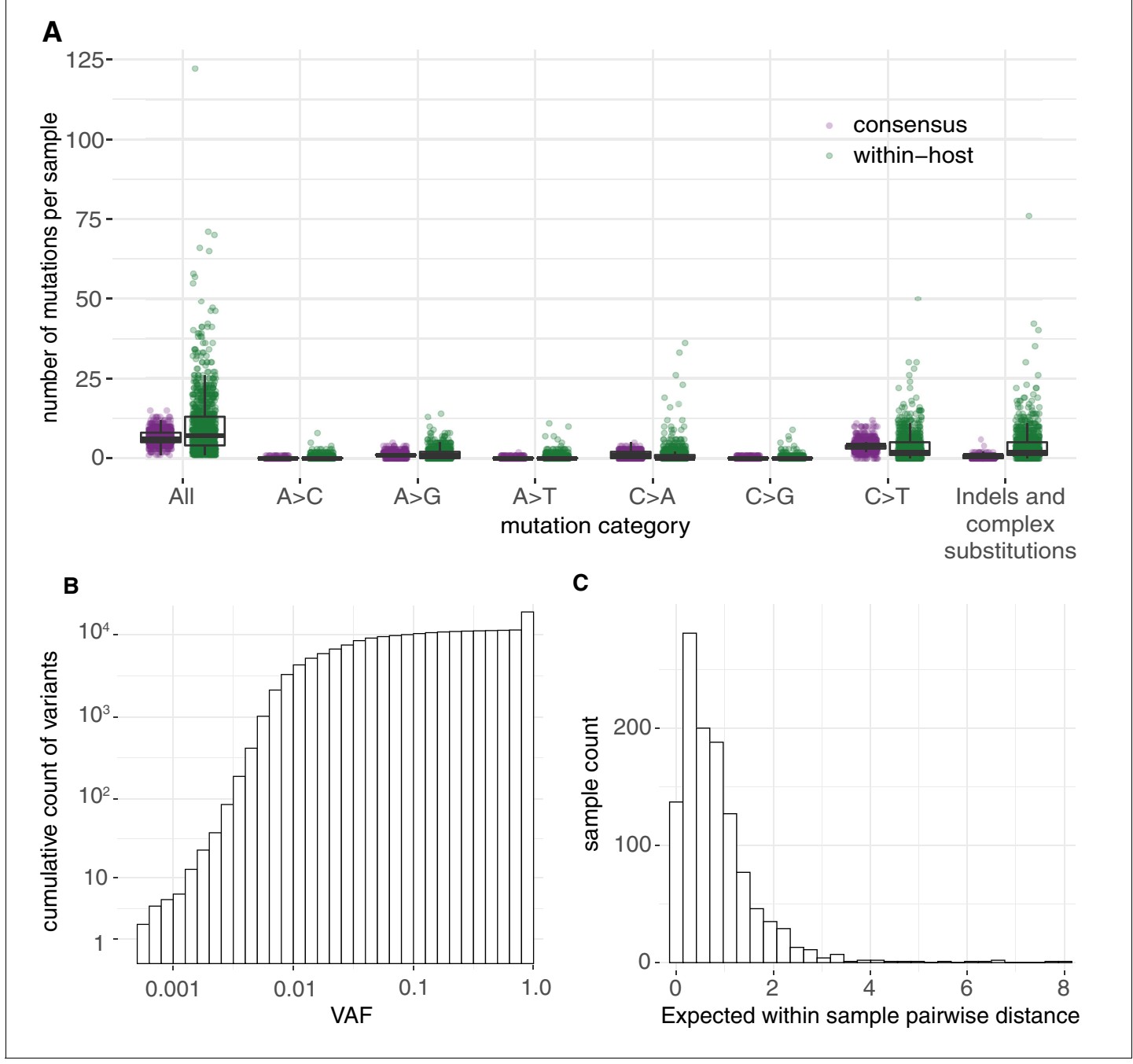

**Figure 1.** Allele frequencies and mutation burden. (A) Number of variants per sample (y-axis) for each mutation type assuming the reference genome as the ancestral allele. (B) A cumulative histogram of the VAFs of all mutation calls. Note that variants shared across samples are counted multiple times and that the 7672 consensus variants correspond to 1079 different changes in 1063 different sites. (C) Histogram of the expected number of mutations separating two randomly sampled genomes for each sample (Materials and methods).

The online version of this article includes the following figure supplement(s) for figure 1:

**Figure supplement 1.** Barplots indicating the mean sequencing depth across the SARS-CoV-2 reference genome for the two replicate runs of the 1181 samples.

**Figure supplement 2.** Dot plots indicating the concordance between variant allele frequency estimates across sequencing replicates in four samples.

**Figure supplement 3.** Estimated overdispersion of variant frequencies and the distribution of sample Ct values.

**Figure supplement 4.** The distribution of the number of variable sites in coding regions among different coding positions.

## The mutational spectra reveal strong strand asymmetries

Analysis of the mutational spectrum can yield insights into the dominant mutational mechanisms underlying the evolution of SARS-CoV-2 during the pandemic. Consistent with previous reports (*Popa et al., 2020*), the mutational spectrum of within-host variant closely resembles that of consensus variants (*Figure 2*). The spectrum shows two striking features: a dominance of C>U and G>U changes with weak extended sequence context, and a large asymmetry between the plus and minus strands, inferred from the high C>U/G>A and G>U/C>A ratios when mutations are mapped to the reference (plus) strand as has previously been described in analyses of SARS-CoV-2 consensus genomes (*Kustin and Stern, 2021*; *Simmonds, 2020*). C>U mutations account for 47.9% of all within-host point mutations compared to 5.3% for G>A mutations (C>U mutations in the minus strand), and G>U mutations account for 14.3% of mutations compared to 1.9% of C>A mutations (Materials and methods). Normalised for the genome sequence composition, the plus/minus strand ratios of the rates of C>U and G>U are 9.6-fold and 7-fold, respectively.

These asymmetries appear difficult to explain whether mutations were direct replication errors. Since any given viral RNA molecule has undergone the same number of plus-to-minus and minus-to-plus replication steps, replication errors are only expected to cause these asymmetries if both steps have very different error rates. For example, if the polymerase had a high C>U error rate in both strands, we would expect a symmetric number of C>U and G>A mutations in the plus strand, as C>Us would be introduced at equal rates when copying the plus or the minus strands (*Figure 2E*). Within a cell, the fact that minus templates are copied many times could theoretically lead to a viral population with fewer G>A mutations but at higher frequencies (*Sawicki et al., 2007*; *V'kovski et al., 2021*). However, the fact that the same asymmetries are observed for consensus variants, which typically represent the fixation of a single genotype, makes this an unlikely factor. Thus, unless the error rates of the RNA-dependent RNA-polymerase in SARS-CoV-2 are very different on both strands, which may be possible given its multisubunit structure (*Zhao et al., 2020*), replication errors seem unlikely to explain the observed asymmetries.

The strong strand asymmetries observed may be more consistent with RNA damage or RNA editing of the plus strand dominating the accumulation of mutations in SARS-CoV-2. The plus strand is the infectious genome, exported from the replication organelles, where minus molecules reside, into the cytoplasm (*Wolff et al., 2020*), translated, packaged into particles, and transmitted between cells and hosts. The plus strand is also present within cells in much larger numbers than the minus strand (*Sawicki et al., 2007*). Thus, the plus strand may be expected to accumulate higher rates of damage or editing, which would manifest as strand asymmetries. If we accept this hypothesis, the dominance of C>U over G>A and G>U over C>A, when mapped to the plus strand, suggests that the dominant forms of RNA editing or damage are C>U and G>U.

RNA-editing enzymes in human cells are able to mutagenise single-stranded DNA and RNA molecules and are known anti-viral mechanisms (*Hoopes et al., 2016*; *Di Giorgio et al., 2020*). Two families of RNA-editing enzymes in particular have been speculated to contribute to the mutational spectrum of SARS-CoV-2, APOBEC cytosine deaminase enzymes causing cytosine to uracil transitions and ADAR enzymes causing adenosine-to-inosine changes (A>G/U>C mutations) (*Di Giorgio et al., 2020*; *Simmonds, 2020*). While we see a high rate of C>U changes in the SARS-CoV-2 spectra, we see much lower rates of A>G/U>C mutations than previously suggested (*Shen et al., 2020*). While the mutational spectrum induced by all APOBEC enzymes is not fully understood, the activity of the better-understood APOBEC3A and 3B enzymes in human cancers is distinctly characterised by a strong sequence context, leading to C>T and C>G changes almost solely at TpC sites (*Alexandrov et al., 2020*). This contrasts with the weak context dependence observed in the SARS-CoV-2 spectrum. While RNA-editing enzymes may contribute to SARS-CoV-2 mutagenesis, direct damage of cytosine and guanine bases could also be consistent with the observed spectrum. For example, spontaneous cytosine deamination would cause C>U transitions while oxidation of guanine bases could explain G>U transversions (*Krokan et al., 2002*). Both forms of damage are common mutagenic processes on DNA in human cells (*Helleday et al., 2014*).

Having described the mutational spectrum, we can also derive approximate estimates of the average diversity within a sample in terms of the expected number of differences between any two genomes found within a sample. Since the allele frequency of a mutation represents the fraction of viral RNA molecules in a sample carrying a mutation, we can estimate the expected mean number of

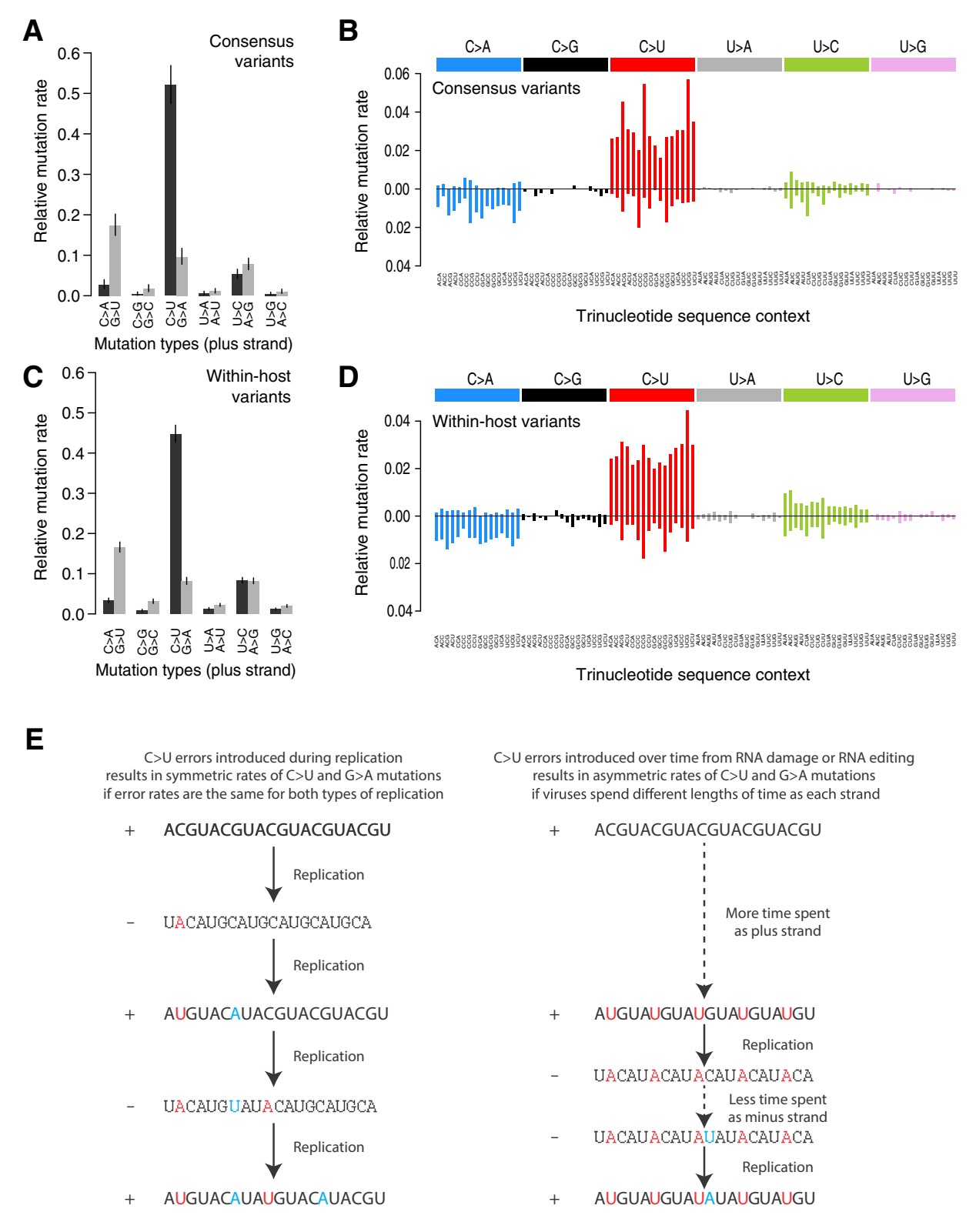

**Figure 2.** Mutational spectra. (**A, C**) Mutational spectra (without sequence context) of consensus (**A**) and within-host (**C**) variants, as mapped to the reference strand and normalised for the composition of nucleotides in the reference genome (MN908947.3). Rates reflect the fraction of the total number of mutations observed. Asymmetries suggest different mutation rates in the plus and minus strands. Error bars depict Poisson 95% confidence intervals. (**B, D**) Mutational spectra in a 96-trinucleotide context of consensus (**B**) and within-host (**D**) variants, as in *Alexandrov et al., 2013*. Mutations

*Figure 2 continued on next page*

*Figure 2 continued*

are represented as mapped to the pyrimidine base, depicted above the horizontal line if the pyrimidine base is in the reference (plus) strand and below it if the pyrimidine base is in the minus strand. Within-host variants observed across more than one sample can represent a single ancestral event or multiple independent events. To prevent highly recurrent events from distorting the spectrum, within-host variants observed across multiple samples were counted a maximum of five times in (C, D). (E) A diagram illustrating how asymmetrical mutation rates of C>U and G>A could be driven by viral sequences spending a longer time as plus strand molecules.

The online version of this article includes the following figure supplement(s) for figure 2:

**Figure supplement 1.** The mutational spectra in a 96-trinucleotide context of recurrent within-host variants.

differences between two genomes from the same host as $2\sum_i p_i(1 - p_i)$ where $p_i$ is the frequency of a variants at position $i$ in the genome. Importantly, this is an approximation as it assumes that each RNA molecule sequenced originated from the entire viral genome, which is not always the case given the abundance of viral sgRNAs (*Wölfel et al., 2020*; *Perera et al., 2020*). These estimates are also likely to be a conservative lower bound as they only include mutations at detectable allele fractions in both replicates. Across samples, we found the expected number of differences between any two genomes within a sample to be 0.83 (*Figure 1C*). Phylogenetic studies estimate a mutation rate in SARS-CoV-2 of 0.001 mutations/bp/year (*Fauver et al., 2020*) or 0.082 mutations/genome/day. Thus, the observed within-host mutation load would be consistent with the expected acquisition of mutations during a relatively short span of several days.

To investigate the possible accumulation of de novo mutations during the course of an infection, we studied 43 individuals for whom we had multiple samples collected at different timepoints (*Figure 3A*, *Figure 3—figure supplement 1*). Overall, the number of within-host variants tended to increase over time and this trend was significant using a Poisson mixed model to control for host-specific effects (p=0.007; *Figure 3B*). To put this finding in context, there was considerable variation in the observed number of within-host variants among samples from the same individual, even if taken on the same day. This could be due to the bottleneck caused by the different sampling methods, which included sputum, swabs, and bronchoalveolar lavage in addition to variation introduced by samples with high Ct values (*Figure 3—figure supplement 2*). High variability between longitudinal samples has also been observed in six out of nine hospitalised patients in Austria (*Popa et al., 2020*). The authors observed multiple instances of the fixation of a variant over the course of an infection. In contrast, we observed no change in the consensus genome sequence in any of the individuals from whom multiple samples were collected. It is possible that the accumulation of within-host mutations could in future be used to estimate time since infection. Indeed, estimates of within-host diversity have previously been used successfully to estimate the time since infection in HIV (*Giorgi et al., 2010*; *Swiss HIV Cohort Study et al., 2011*; *Puller et al., 2017*). However, determining if a consistent signal could be observed given the extensive variation observed between repeated samples on the same day would require analysis of time-series data in a larger number of individuals with high-quality epidemiological data.

## Within-host variants display strong purifying selection against nonsense mutations

To study the extent of selection acting on within-host variants and consensus variants, we calculated dN/dS ratios using the dNdScv package (Materials and methods). Most commonly used software to calculate dN/dS use simple substitution models (often using a single transition–transversion ratio), which can lead to considerable biases and false signals of selection under neutrality (*Martincorena et al., 2017*; *Van den Eynden and Larsson, 2017*). dNdScv uses a Poisson framework allowing for complex substitution models including context dependence, strand asymmetry, non-equilibrium sequence composition, and estimation of dN/dS ratios for missense and nonsense mutations separately. This model is particularly suitable for datasets with low mutation density in lowly recombining or non-recombining populations, as it is the case in SARS-CoV-2 genomic data *Martincorena et al., 2017*.

As expected, dN/dS ratios for consensus variants are under clear purifying selection (*Figure 4A*), with particularly strong selection against nonsense mutations. This is similar to that observed in other RNA viruses such as Influenza *Xue and Bloom, 2020*. Within-host variants reaching moderately high

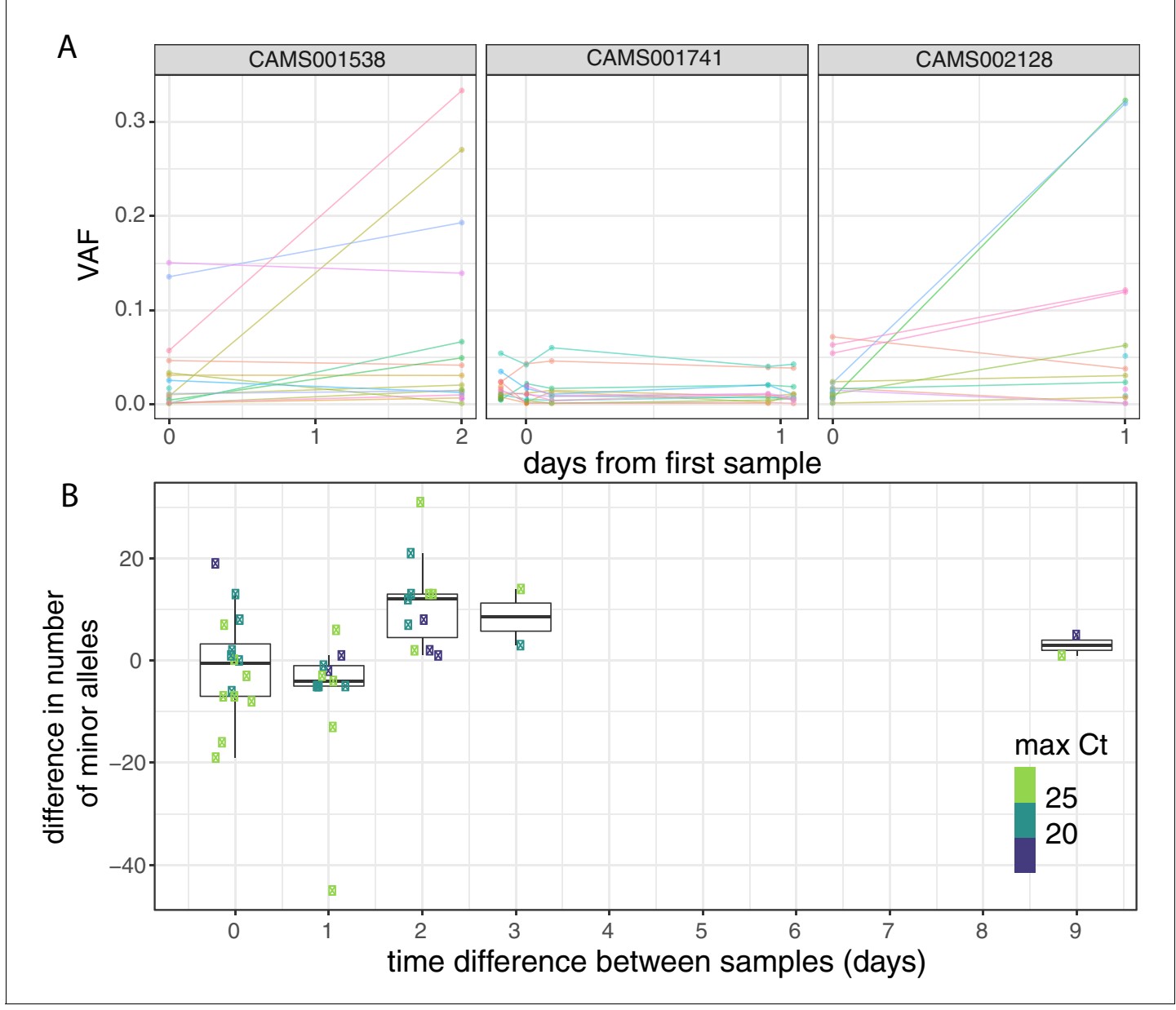

**Figure 3.** Longitudinal differences in within-host variant frequencies. (A) Frequencies of within-host variants for three selected hosts where multiple samples were taken over consecutive days. Samples taken on the same day have been offset by a small distance. Plots for all hosts with multiple samples are given in *Figure 3—figure supplement 1*. (B) The difference in the number of within-host variants between pairwise combinations of samples taken from the same host. The order for samples taken on the same day was randomised, and the colour of the point indicates the maximum of the two Ct values for the corresponding samples.

The online version of this article includes the following figure supplement(s) for figure 3:

**Figure supplement 1.** Frequencies of within-host variants for all hosts where multiple samples were taken over consecutive days.

**Figure supplement 2.** Proportion of shared variants between each pair of samples taken from the same host on the same day.

allele frequencies (VAF > 10%) display similarly strong purifying selection against missense and nonsense mutations as consensus variants (*Figure 4A*), while within-host variants at low allele frequencies appear to be under more relaxed purifying selection, as it may be expected. Purifying selection against nonsense mutations can also be observed at the level of allele frequencies (*Figure 4B*). Overall, the similarity of dN/dS ratios for consensus and moderate-VAF within-host variants suggests that

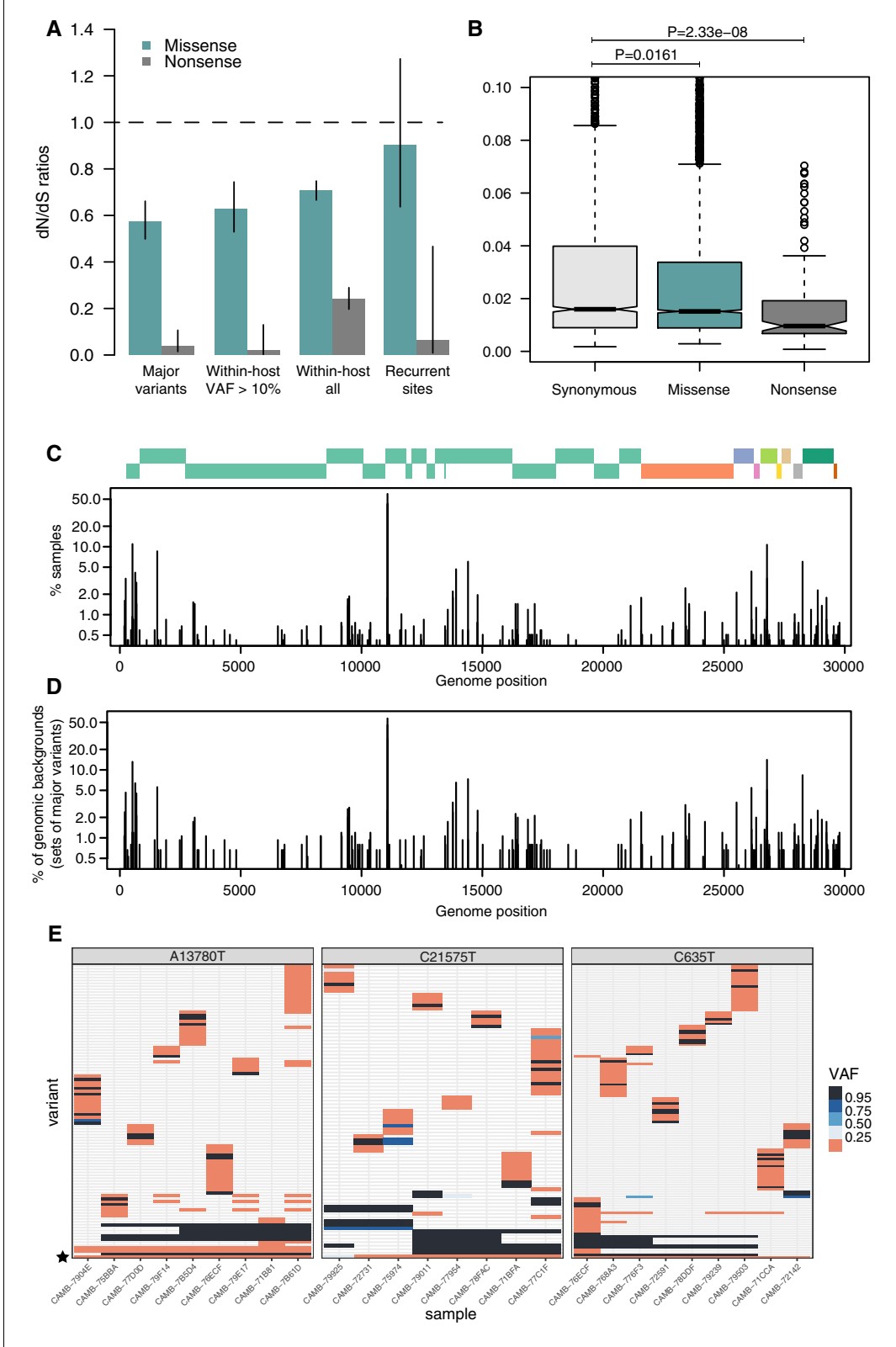

**Figure 4.** Patterns of selection and recurrent within-host variants. (**A**) Genome-wide dN/dS ratios for missense and nonsense mutations (Materials and methods). Error bars depict 95% confidence intervals from the Poisson maximum-likelihood model. (**B**) ≥VAFs of within-host variants as a function of their predicted coding impact. p-values were calculated with Wilcoxon tests. (**C**) The top panel depicts the coordinates of the annotated peptides in the reference genome, coloured according to their ORF. The bottom panel depicts the frequency at which recurrent within-host variants (defined as

*Figure 4 continued on next page*

*Figure 4 continued*

those seeing in five or more samples) occur in the dataset. (D) Frequency of recurrent within-host variants (as in C) across different genomic backgrounds in the dataset (defined as the set of consensus variants in the sample). (E) Heatmaps of variant allele frequencies in samples containing three common within-host variants found at potential mutational hotspots are shown. The diversity of consensus variants with VAF $\geq$ 95% (black tiles) across samples is better explained by independent acquisitions of the minority variant rather than transmission.

selection within hosts, rather than during transmission, may explain a considerable fraction of the extent of purifying selection observed in consensus sequences.

## Many recurrently mutated sites appear to represent mutational hotspots

Some sites across the SARS-CoV-2 genome appear to have mutated independently multiple times, resulting in homoplasic sites across the viral phylogeny (*Nicola et al., 2020*). Some of these sites have also been reported to recurrently appear as within-host variants, although it remains unclear to what extent these events represent recurrent sequencing errors, contamination between samples, co-infection of a sample by multiple lineages, recombination, mutational hotspots or convergent positive selection (*Lythgoe et al., 2020*; *Nicola et al., 2020*; *Popa et al., 2020*). Co-infection by multiple lineages could possibly arise from superinfection, where an individual acquires infection from multiple sources, or by co-transmission of multiple lineages from host to host following an episode of superinfection.

*Figure 4E* represents the distribution of recurrent within-host variants including positions where the reference allele is found in the minority. Within-host variation is observed in at least 5 (0.4%) of our samples at 215 sites within the SARS-CoV-2 genome (*Supplementary file 4*). Sequencing or PCR errors are unlikely to contribute substantially to the recurrent variants observed in our dataset thanks to the use of replicates and the ShearwaterML algorithm. One mechanism by which the same within-host variant can be observed across multiple closely related samples is transmission of within-host variants between contacts, when a population of virions is transmitted between hosts. Under this scenario, sharing of within-host variants will be expected between samples with identical fixed variants at sites which do not exhibiting within-host variation. The preservation of a within-host variant is incompatible with the simultaneous fixation of new variants (the within-host allele would either be purged or hitchhike and become fixed). Instead, we find that most recurrent within-host variants occur across lineages, independently of the genomic background of fixed variants present in a sample (*Figure 4E*). This pattern is suggestive of recurrent mutation, although it could also be consistent with complex histories of superinfection and recombination.

If we accept the hypothesis of recurrent mutation, there remains the question of whether this is caused by mutational hotspots or convergent positive selection. We observed that 28.5% (57/200) of recurrent ($n \geq 5$) within-host variants are predicted to cause synonymous mutations and estimates of the dN/dS ratio corrected for the trinucleotide sequence context indicate that, as a group, these recurrent variants are under some purifying selection (*Figure 4A*). This suggests that most recurrent within-host variants are likely to represent hypermutable sites rather than convergent positive selection. Their mutational spectrum suggests an enrichment for C>U changes, particularly at sites preceded by a pyrimidine, but the mechanisms behind their apparent hypermutability remain unclear (*Figure 2—figure supplement 1*). Still, this analysis does not rule out the possibility that a minority of recurrent within-host variants are the result of convergent positive selection. Indeed, a number of recurrent within-host variants are also observed to frequently occur at the consensus level in the broader COG-UK dataset (*Supplementary file 4*). A plausible example of positive selection is the spike glycoprotein mutation D614G, which was rapidly increasing in frequency throughout the world at the time these samples were collected: within-host variation at this position appears in 27 of our samples, with the D614G allele often present at high allele frequencies.

## Extensive sharing of minority variants across a diverse genomic background suggests caution is needed when using within-host variants for the inference of transmission

In order to investigate the genetic background of our samples, we generated a maximum-likelihood phylogeny of all consensus genomes produced by the COG-UK consortium by the end of May 2020,

including those for the samples on our dataset (*Figure 5A*). This showed that our samples represent a broad range of the SARS-CoV-2 genetic diversity found in the UK at that time and that the diversity observed among our samples was not primarily driven by geographical location.

To explore the relationship between within-host variants and the consensus phylogeny, we identified within-host variants that are shared between samples, as illustrated by links between the tips of the phylogenetic tree in *Figure 5D*. This confirmed that within-host variants are often shared between samples that are distant on the consensus phylogeny. Both a high level of recombination and a large transmission bottleneck would be required for within-host variants to be maintained across long transmission chains in order to explain the simultaneous preservation of some within-host variants with the fixation of others. While recent studies have suggested that the bottleneck in SARS-CoV-2 transmission can be on the order of $10^2$–$10^3$ virions, they also identified substantial overlaps in the fraction of shared minority variants between samples unrelated by close transmission (*Popa et al., 2020*; *Mara and Chu, 2020*). The sharing of within-host variants between consensus genomes as divergent as 10 SNPs, indicating multiple months of separation between the samples suggests that a more likely explanation is that many of these shared within-host variants are the result of recurrent mutation or co-infection (*Figure 6D*). Indeed, a re-analysis of the Austrian data estimated much smaller bottleneck on the order of 1–3 virions (*Martin Michael and Katia, 2021*).

To further investigate the correspondence between shared within-host variants and transmission, we used the transcluster algorithm to infer the probability distribution of the number of intermediate hosts that separate each pair of samples (*Stimson et al., 2019*). This pairwise approach accounts for the serial interval of the virus as well as its evolutionary rate using the difference in the number of SNPs between each pair of consensus genomes. *Figure 6B* illustrates the relationship between the

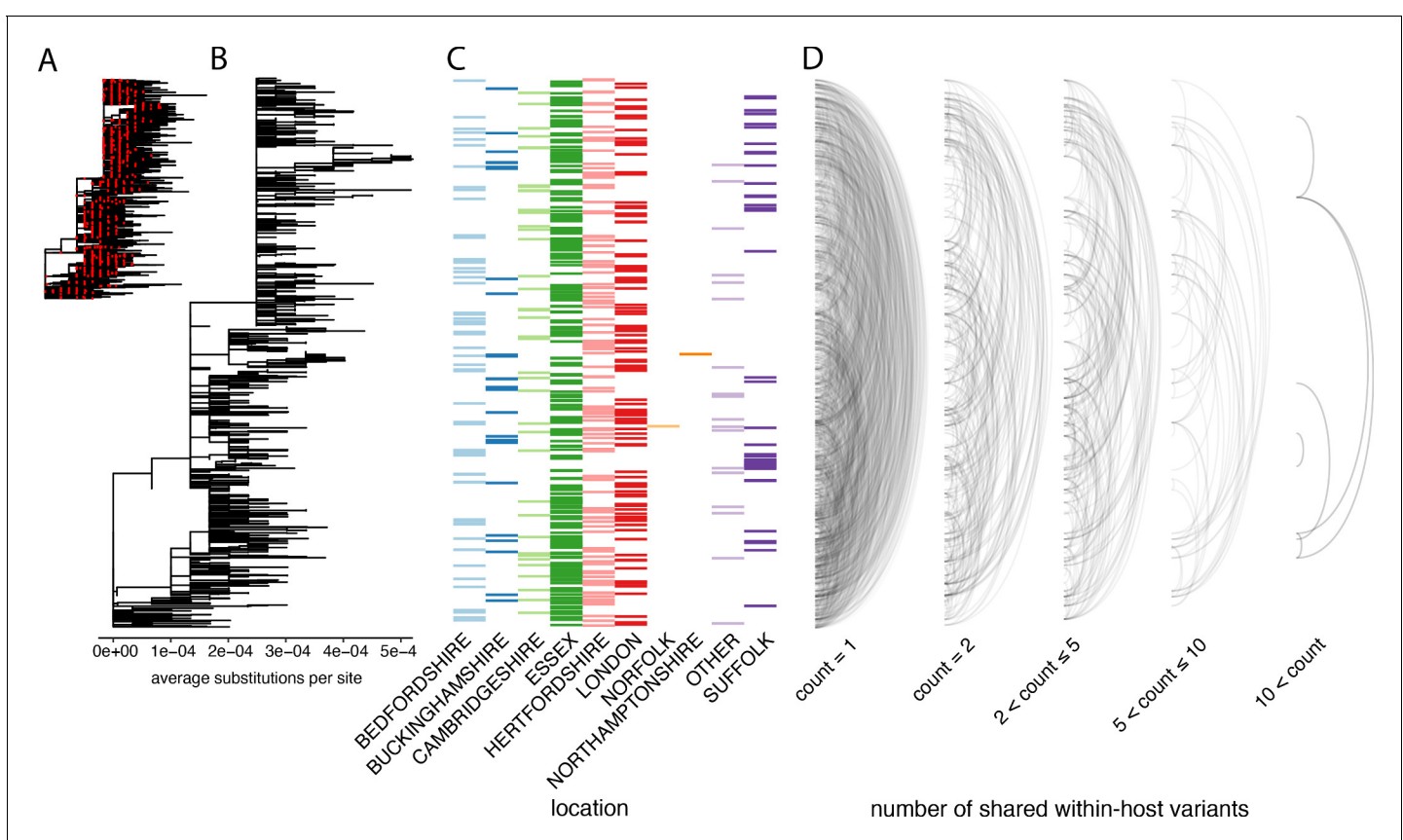

**Figure 5.** The distribution of shared within-host variants between samples with respect to the inferred consensus phylogeny. (A) A maximum-likelihood phylogeny of all COG-UK consensus genomes available on 29 May 2020. Red dots indicate the location of those samples that were deep sequenced in replicate. (B) The same phylogeny restricted to those samples taken for deep sequencing. (C) The region each patient's home address was located. (D) Links are drawn between tips of the phylogeny that share within-host minority. Links restricted to those variants seen in less than 2% of individuals and are separated based on the number of variants shared between samples.

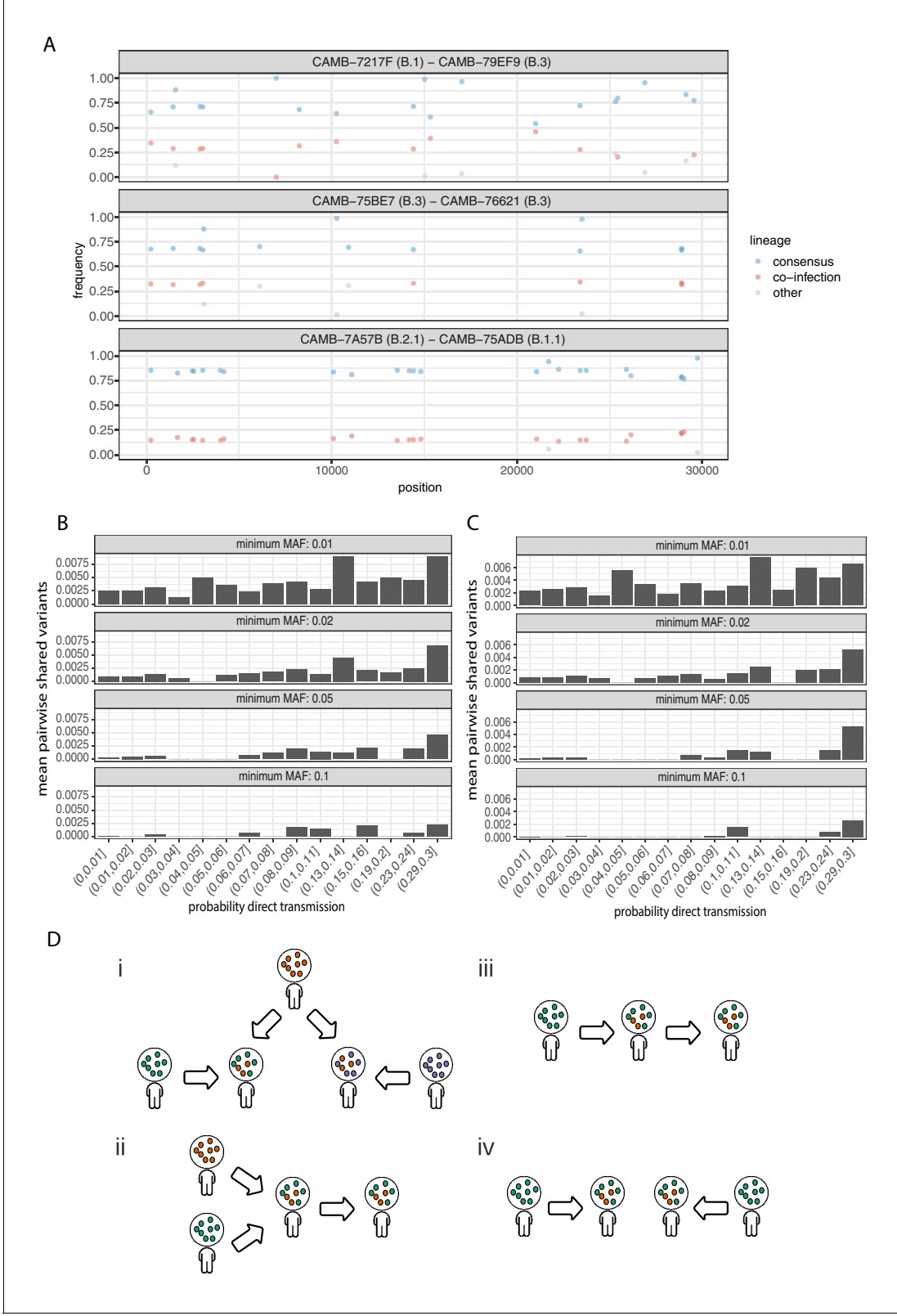

**Figure 6.** Potential mixed infections and the relationship between transmission and shared within-host variants. (**A**) An example of three samples identified as potential mixtures. The consensus lineage is given first and coloured blue, while the potentially co-infecting lineage is given second and coloured red. Minority variants that do not match the co-infecting lineage are coloured grey. (**B**) The mean number of shared iSNVs shared by each pair of samples binned by the probability they were the result of a direct transmission according to the model of *Stimson et al., 2019*. Results, with a

*Figure 6 continued on next page*

*Figure 6 continued*

minimum minor allele frequency of 0.01, 0.02, 0.05, and 0.1 are shown in each of the facets. Within-host variants observed in more than 2% of samples were excluded. (C) The same plot as *Figure 3B* but having removed all samples that were inferred to be mixed infections. (D) A diagram demonstrating the four scenarios that can lead to shared within-host variants. (i) Superinfection of a common strain. (ii) Superinfection followed by co-transmission (iii) Transmission of the within-host variants through a large bottleneck. (iv) Independent de novo acquisitions of the same within-host variants. Shared within-host variants in scenarios (ii, iii) are concordant with the transmission tree, while (i, iv) are discordant, potentially confounding transmission inference efforts.

The online version of this article includes the following figure supplement(s) for figure 6:

**Figure supplement 1.** All samples identified as potential mixtures.

mean number of shared within-host variants and the probability that the respective infections were a result of direct transmission. A weak correlation with transmission probability was found for shared within-host variants of high allele frequency, but not for those of low allele frequency.

## Co-infections correlate with the consensus phylogeny consistent with cases of superinfection induced by population structure

It had been proposed that co-infection by different lineages was a significant source of within-host variation in SARS-CoV-2 although this was later amended (*Lythgoe et al., 2020*). Instances of co-infection are common in other RNA viruses including enterovirus and HIV (*Dyrdak et al., 2019*; *Liu et al., 2002*). Co-infection by multiple lineages would result in the allele frequencies of divergent bases to appear as within-host variants and reflect the proportions of either haplotype. Detecting instances of co-infection is important in the context of both transmission and to accurately characterise the within-host mutation rate (*Chris et al., 2017*; *Cudini et al., 2019*). To identify possible co-infected samples, we compared the site frequency spectrum for each of our samples with all of the consensus genomes in the COG-UK dataset at the end of May 2020. A linear model was used to identify mixtures of two consensus genomes that could explain the allele frequencies of variants within a sample better than a single consensus genome. Mixtures were considered if two consensus genomes could explain at least two additional within-host variants over that of a single consensus genome without including more than one variant not found within the sample. Samples identified by the model as being a possible mixture of consensus genomes were then visually inspected to determine whether the putative co-infections were convincing. This resulted in 36 putative co-infected samples, with a representative example shown in *Figure 6A*. The frequencies of within-host variants of all 36 samples, along with consensus genomes that comprise the putative mixture, are shown in *Figure 6—figure supplement 1*. To determine the potential impact of co-infections on transmission inference, we excluded all 36 samples and re-ran the pairwise transmission analysis using the transcluster algorithm on the remaining samples. *Figure 6C* indicates that the remaining within-host variants no longer correlate as well with the probability of direct transmission, suggesting that much of the correlation observed previously was potentially driven by co-infections.

It had been suggested that correlations between co-infections and the consensus phylogeny could be driven by the co-transmission of multiple strains (*Lythgoe et al., 2021*). However, as the transmission signal in the minority variants in the remaining samples has reduced, it is likely that in some instances co-infection of certain strains is driven by multiple episodes of superinfection (co-infection from two different infection sources). As these samples and those from *Lythgoe et al., 2021* were acquired from hospitalised patients, the correspondence with the consensus phylogeny could be driven by structured superinfection within hospital COVID-19 wards. The prevalence of infection within hospitals, particularly during the first 'wave' of the COVID-19 pandemic was often considerably higher than in the community with up to 50% of the consultant A and E workforce at one Welsh hospital testing positive in April 2020 (*Black et al., 2020*). This contrasts with the finding that approximately 6% of the UK population had been infected with SARS-CoV-2 by the end of June (*Ward et al., 2020*). Within-hospital transmission was also common in the early stages of the outbreak in Wuhan, with 41% of 138 patients found to have contracted SARS-CoV-2 in hospital (*Wang et al., 2020*). While it is not possible to rule out cross-contamination between samples, this contamination would have to be structured (e.g. occurring within the hospital rather than at a sequencing centre) to explain the correlations between the inferred co-infections and the consensus

phylogeny. Batch effects were carefully controlled for by Lythgoe et al., and as we identified a similar signal using an independent dataset with samples sequenced in replicate, structured superinfection provides a more likely explanation. In cases of higher prevalence, repeated episodes of superinfection rather than co-transmission could complicate the use of within-host minority variants in transmission inference (*Figure 6D*).

## Discussion

We find a considerable amount of genetic diversity within individual SARS-CoV-2 infections that cannot be explained by technical artefacts. This is consistent with the quasispecies population structure typically found in RNA viruses (*Vignuzzi et al., 2006*; *Eigen, 1993*), including related betacoronaviruses SARS-CoV-1 and MERS (*Lu et al., 2017*; *Xu et al., 2004*). By analysing the frequency of variants within individual samples, we estimate that two randomly chosen genomes within a sample differ by 0.83 variants on average. Since most of these samples were probably collected more than a week after the time of infection, and since the SARS-CoV-2 genome is known to acquire approximately one mutation every 2 weeks, these findings are broadly consistent with the hypothesis that within-host variation is largely due to the accumulation of de novo mutations within the host or, at least, in the span of a few days. Further support for this hypothesis comes from our analysis of hosts sampled at multiple timepoints, showing that the number of within-host variants tends to increase during the course of an infection. Increased numbers of within-host variants have also been observed in immunocompromised patients: consistent with the acquisition of de novo mutations within the host (*Siqueira Juliana et al., 2020*).

These data show that within-host variants have a similar mutational spectrum to the consensus variants that define between-host variation. Both are characterised by clear purifying selection, as would be expected if virions with disadvantageous mutations failed to survive or propagate within the host. Strikingly, the mutational spectrum of SARS-CoV-2 exhibits a near complete asymmetry between the plus and minus strands and is dominated by C>U and G>U mutations. This seems consistent with RNA damage or RNA editing of the plus strand. Whilst current knowledge of the mutational spectrum of APOBEC enzymes on RNA molecules is insufficient to assess the likelihood that they actively contribute to SARS-CoV-2 evolution, we note that the spectrum observed is very different to that of APOBEC mutagenesis in DNA from human cells. Instead, direct damage of cytosine and guanine bases could also be consistent with the observed spectrum of C>U and G>U changes in SARS-CoV-2. Many of the within-host variants that we have identified are shared between infected individuals located on distant branches of the consensus phylogeny, and this is congruent with previous reports that the SARS-CoV-2 consensus phylogeny has many homoplasies that also appear as within-host variants (*Lythgoe et al., 2021*). It appears likely that this is largely due to recurrent mutation although co-infection between lineages could also be a relevant factor in high transmission settings. The vast majority of our samples appear to comprise a single lineage, but we find evidence of putative co-infection by multiple lineages in approximately 3–4% of samples, which is likely an underestimate of the extent of co-infection in our dataset.

In other viral and bacterial diseases, within-host variants provide a valuable source of information for the inference of transmission chains (*De Maio et al., 2018*; *Chris et al., 2017*; *Worby et al., 2017*; *Arias et al., 2016*) and have been shown to improve the accuracy of inferences based on consensus genomes (*Worby et al., 2014*; *Hall et al., 2015*; *De Maio et al., 2016*; *Campbell et al., 2018*). In the case of SARS-CoV-2, initial estimates suggested that the transmission bottleneck may be large (*Lythgoe et al., 2020*; *Popa et al., 2020*). However, revised estimates suggest that a much smaller bottleneck is likely similar to that seen in Influenza (*Lythgoe et al., 2021*). We find some evidence of a correlation between sharing of within-host variants and the probability of direct transmission, a signal that we find is partially driven by co-infection, but the current dataset lacks the epidemiological resolution to evaluate the utility of this in reconstructing transmission chains, which might depend on the prevalence of infection and other local circumstances such as superinfection within COVID-19 hospital wards.

In summary, these data show that there is considerable genetic diversity in the SARS-CoV-2 viral populations within individual hosts. Within-host diversity seems consistent with the accumulation of de novo mutations during the course of infection but can also result from co-infection with different lineages. Analysis of within-host variation could potentially be used to improve the inference of

transmission chains, but this approach requires caution because of the confounding effects of recurrent mutation and unrelated episodes of superinfection. More detailed studies are required to evaluate the transmission bottlenecks that govern the propagation of within-host variants from host to host and to examine the patterns of within-host variation associated with different epidemiological circumstances.

## Materials and methods

### Sample selection and ethics

The 1181 samples were taken as a random subset from a larger prospective study into SARS-CoV-2 infections at Cambridge University Hospitals National Health Service Foundation Trust (CUH; Cambridge, UK), a secondary care provider and tertiary referral centre in the East of England (*Meredith et al., 2020*; *Hamilton et al., 2020*). This study was done as part of surveillance for COVID-19 under the auspices of Section 251 of the National Health Service Act 2006. It therefore did not require individual patient consent or ethical approval. The COVID-19 Genomics UK (COG-UK) study protocol was approved by the Public Health England Research Ethics Governance Group.

### Sequencing

A single swab was taken for each sample. Two libraries were then generated from two aliquots of each sample with separate RT, PCR amplification, and library preparation steps in order to evaluate the quality and reproducibility of within-host variant calls. The ARTIC protocol v3 was used for library preparation (a full description of the protocol used is available at http://dx.doi.org/10.17504/protocols.io.be3wjgpe).

### Alignment and variant calling

Alignment was performed using the ARTIC Illumina nextflow pipeline available from https://github.com/connor-lab/ncov2019-artic-nf (*Bull, 2020*; copy archived at swh:1:rev:8af5152cf7107-c3a369b996f5bad3473d329050c). The pipeline trims primer and adapter sequences prior to alignment to the reference genome using bwa (MN908947.3) (*Li and Durbin, 2009*). Variant calling was performed using ShearwaterML, and bam2R was used to count the number of reads supporting each base at each site. Both these methods are available as part of the deepSNV R package (*Gerstung et al., 2014*; *Martincorena et al., 2015*). In order to create a base-specific error model for the SARS-CoV-2 genome, we randomly selected sequencing data from 100 samples (50 from each set of duplicates) from the 468 samples that met the following criteria: (1) $\rho$ value from beta-binomial model $\leq 0.02$; (2) proportion of genome with at least $500\times$ coverage $\geq 0.9$ for both duplicates; (3) absolute difference in median coverage between replicates $\leq 20,000$; and (4) only one sample sequenced from a given donor. For each sample, non-reference sites with VAF $\geq 0.01$ were set as uninformative (i.e. depth for all base types were set to 0) in order to enable calling of recurrent, high VAF mutations. However, two sites (11074 and 25202) were effectively excluded from variant calling as they were set as uninformative for all samples in the normal panel.

Variants were called separately for each set of duplicates. The initial ShearwaterML calls were filtered using the following criteria: (1) total depth at variant site $\geq 100\times$ and (2) Benjamini–Hochberg false discovery rate q-value $\leq 0.05$ in one duplicate and unadjusted p-value $\leq 0.01$ in the other duplicate. The adjustment of p-values was performed by considering the five mutation types (three non-reference bases, insertions, and deletions) at all sites with $\geq 100\times$ coverage in the 1181 samples from a given duplicate set. Variants at consecutive sites were merged into single events if the difference in VAFs $\leq 0.05$. The VAF for variants that were called using the ShearwaterML algorithm in both replicates was calculated as $\frac{x_{1,j}+x_{2,j}}{n_{1,j}+n_{2,j}}$ where $x_{1,j}$ and $x_{2,j}$ are the alternative (non-reference) allele counts of variable site $j$ in replicates 1 and 2 of a given sample, and $n_{1,j}$ and $n_{2,j}$ are the local read counts at the respective sites. All scripts used to perform the variant calling are available in the accompanying GitHub repository.

## Variant summary statistics

The mean standard deviation in VAF was calculated assuming a Bernoulli distribution at each site, such that given a variant at frequency $p_i$, the standard deviation was calculated as $\sqrt{p_i(1-p_i)}$. The average over all variable sites was then taken. The expected number of differences between two randomly chosen genomes within a sample was calculated as $2\sum_i p_i(1-p_i)$.

## Beta-binomial modelling of replicate samples

To quantify the level of discordance in the allele frequencies between technical replicates, for each pair replicates we first identified a set of variable sites, which are expected to contain both artefacts and within-host mutations. All non-reference variable sites with coverage in both replicates >100x and mean VAF from both replicates higher than 1% and lower than 90% were considered for the beta-binomial modelling. In Illumina sequencing protocols where adapters are ligated before amplification and PCR duplicates can be removed computationally, VAFs may be expected to show binomial variation between technical replicates. However, amplicon sequencing can lead to preferential amplification of some molecules leading to higher variation in VAFs. We quantified the extent of variation above binomial sampling in the VAFs of variable sites between replicates by fitting a beta-binomial model to the allele counts of variable sites, obtaining a maximum-likelihood estimate of the ρ (overdispersion) parameter for each pair of replicate samples. Let $x_{1,j}$ and $x_{2,j}$ be the alternative (non-reference) allele counts of variable site $j$ in replicates 1 and 2 of a given sample, and $n_{1,j}$ and $n_{2,j}$ be the local coverages at the site. We can calculate an approximate likelihood for the beta-binomial model using the following equation, where $P$ depicts the beta-binomial density function:

$$L = \prod_{i\in\{1,2\}} \prod_{j} P\left(x_{i,j}, n_{i,j}, p=\frac{x_{1,j}+x_{2,j}}{n_{1,j}+n_{2,j}}, \rho\right)$$

A maximum-likelihood estimate for the overdispersion parameter was obtained by grid search. The overdispersion parameter, measuring the level of discordance in allele frequencies between the replicates, was found to correlate strongly with the Ct value of the sample.

## Mutational spectrum

The mutational spectra shown in *Figure 2* were generated assuming that the allele in the reference sequence (SARS-CoV-2 isolate Wuhan-Hu-1, MN908947.3) represents the ancestral allele. The normalised mutation rates (r) for each of the 192 possible changes (j) in a trinucleotide context were calculated as:

$$r_j = \frac{n_j}{L_j \sum_j \frac{n_j}{L_j}}$$

where $n_j$ is the total number of mutations observed for a trinucleotide change $j$, and $L_j$ is the total number of times that the corresponding trinucleotide is present in the reference genome (MN908947.3). When observing the same mutation in multiple samples, it is not always straightforward to determine the number of independent mutational events that this represents. For consensus variants, this can be better estimated using a phylogeny, although recurrent hotspots can cause errors in the phylogenetic reconstruction. For simplicity, mutations observed across samples as consensus variants were counted only once for the estimation of $n_j$. For within-host variants, a more relaxed approach was used since, based on our results, multiple instances of the same within-host variant across samples are more likely to represent independent events. Within-host variants observed in more than five samples were only counted five times for the calculation of $n_j$, to avoid a small number of highly recurrent hotspots from dominating the trinucleotide spectrum.

## Selection analysis

Analyses of selection were carried out using the dNdScv package (*Martincorena et al., 2017*). In contrast to traditional implementations of dN/dS developed for divergent sequences, which rely on Markov-chain codon substitution models (*Goldman and Yang, 1994*), dNdScv was developed for comparisons of closely related genomes. Under low divergence rates, such as those in the densely sampled SARS-CoV-2 phylogeny, observed changes typically represent individual mutational events,

which can be modelled using a Poisson framework instead of Markov-chain substitution models. This enables the use of more complex and realistic substitution models, including context dependence, strand asymmetry, non-equilibrium sequence composition, and separate estimation of dN/dS ratios for missense and nonsense mutations. These changes can be important to avoid false signals of negative or positive selection under neutrality when using simplistic substitution models (*Martincorena et al., 2017*). Here we used the default substitution model in dNdScv, which uses 192 rate parameters to model all possible mutations in both strands separately in a trinucleotide context, as well as two ω parameters to estimate dN/dS ratios for missense and nonsense mutations separately. dNdScv files and code needed to generate dN/dS ratios from SARS-CoV-2 data are available at https://github.com/gtonkinhill/SC2_withinhost, DOI: 10.5281/zenodo.5115287.

dN/dS ratios on polymorphism data need to be interpreted with caution. This is both because dN/dS ratios can be time dependent (*Rocha et al., 2006*), providing weaker signals of selection for more recent changes, and because dN/dS ratios can also behave non-monotonically with respect to selection coefficients under idealised conditions of free recombination when using nucleotide diversity within a population (*Kryazhimskiy and Plotkin, 2008*). The former can also result in an excess of non-synonymous changes at deeper branches in the phylogeny due to incomplete purifying selection as has been observed previously in RNA viruses (*Kustin and Stern, 2021*). The latter can result from strong positive selection causing a reduction in the effective population size (and in nucleotide diversity) of non-synonymous sites under free recombination. However, the potential loss of monotonicity should not be a concern in our analyses of SARS-CoV-2 data. This is both because free recombination between adjacent sites is unrealistic for SARS-CoV-2 and because collections of mutations were identified by comparison of derived sequences to an ancestral reference genome, instead of by comparison of two derived sequences at any one time.

## Consensus phylogeny construction

Consensus genomes for each replicate were generated using the ARCTIC SARS-CoV-2 bioinformatics pipeline. A multiple sequence alignment was generated using MAFFTv7.464, and any sites that were discordant between replicates were set to be ambiguous (*Katoh et al., 2002*). Sites that have previously been identified to create difficulties in generating phylogenies were masked using bedtools v2.29.2 using the VCF described in *Nicola et al., 2020*; *Quinlan and Hall, 2010*. Fasttree v2.1.11 was used to generate a maximum-likelihood phylogeny (*Price et al., 2010*).

## Transmission model

To investigate transmission, samples were only considered if both replicates produced high-quality consensus genomes. When multiple samples from the same host were available, the earliest sample was used. Pairwise SNP distances were generated between the consensus genomes using pairsnp v0.2.0 (*Tonkin-Hill, 2018*). The distribution of the underlying number of intermediate transmission events between each pair of samples was then inferred using an implementation of the transcluster algorithm (*Stimson et al., 2019*; *Tonkin-Hill, 2020*). The serial interval and evolutionary rate were set to 5 days and 1e-3 substitutions/site/year (*Fauver et al., 2020*; *Zhang et al., 2020*). Within-host variants observed in more than 2% of samples were excluded from the histograms in *Figure 6B,C*.

## Identification of potential mixed infections

Potential mixed infections were identified using a linear model by testing whether the allele frequencies in a sample could be better explained by the inclusion of an additional consensus genome from the COG-UK dataset of 29 May 2020. Additional sample mixtures were considered if the addition of a COG-UK consensus genome could explain at least 2 iSNVs and have at most one variant that was not found in the alleles of the sample. This identified 54 putative mixtures, which were then screened manually to obtain 36 potentially mixed samples. The code used to run this analysis is available in the supplementary materials.

Analysis data and code available from: https://github.com/gtonkinhill/SC2_withinhost, DOI: 10.5281/zenodo.5115287 (*Tonkin-Hill et al., 2021*; copy archived at swh:1:rev:5a20ba005ddcbe9b81f701a248a2ae348d2d9bd8).

Dehumanised sequence data is available from the ENA under accession number ERP126512.

## Acknowledgements

This work was funded by COG-UK, supported by funding from the Medical Research Council (MRC) part of UK Research and Innovation (UKRI), the National Institute of Health Research (NIHR) and Genome Research Limited, operating as the Wellcome Sanger Institute; the Wellcome Trust (Senior Fellowship to IG ref: 207498/Z/17/Z and PhD Scholarship to GTH ref: 204016/Z/16/Z); the Academy of Medical Sciences and the Health Foundation (Clinician Scientist Fellowship to MET); and the Cambridge NIHR Biomedical Research Centre (MET).

## Additional information

### Funding

| Funder | Grant reference number | Author |
| --- | --- | --- |
| Wellcome Trust | 204016/Z/16/Z | Gerry Tonkin-Hill |
| Wellcome Trust | 207498/Z/17/Z | Ian G Goodfellow |
| COG-UK | | Gerry Tonkin-Hill<br>Inigo Martincorena<br>Roberto Amato<br>Andrew RJ Lawson<br>Moritz Gerstrung<br>Ian Johnston<br>David K Jackson<br>Naomi Park<br>Stefanie V Lensing<br>Michael A Quail<br>Sonia Gonçalves<br>Cristina Ariani<br>Michael Spencer Chapman<br>William L Hamilton<br>Luke W Meredith<br>Grant Hall<br>Aminu S Jahun<br>Yasmin Chaudhry<br>Myra Hosmillo<br>Malte L Pinckert<br>Iliana Georgana<br>Anna Yakovleva<br>Laura G Caller<br>Sarah L Caddy<br>Theresa Feltwell<br>Fahad A Khokhar<br>Charlotte J Houldcroft<br>Martin D Curran<br>Surendra Parmar<br>Alex Alderton<br>Rachel Nelson<br>Ewan M Harrison<br>John Sillitoe<br>Stephen D Bentley<br>Jeffrey C Barrett<br>M Estee Torok<br>Ian G Goodfellow<br>Cordelia Langford<br>Dominic P Kwiatowski |
| Medical Research Council | | Gerry Tonkin-Hill<br>Inigo Martincorena<br>Roberto Amato<br>Andrew RJ Lawson<br>Moritz Gerstrung<br>Ian Johnston<br>David K Jackson<br>Naomi Park<br>Stefanie V Lensing<br>Michael A Quail<br>Sonia Gonçalves<br>Cristina Ariani<br>Michael Spencer Chapman |

| | William L Hamilton |
| | Luke W Meredith |
| | Grant Hall |
| | Aminu S Jahun |
| | Yasmin Chaudhry |
| | Myra Hosmillo |
| | Malte L Pinckert |
| | Iliana Georgana |
| | Anna Yakovleva |
| | Laura G Caller |
| | Sarah L Caddy |
| | Theresa Feltwell |
| | Fahad A Khokhar |
| | Charlotte J Houldcroft |
| | Martin D Curran |
| | Surendra Parmar |
| | Alex Alderton |
| | Rachel Nelson |
| | Ewan M Harrison |
| | John Sillitoe |
| | Stephen D Bentley |
| | Jeffrey C Barrett |
| | M Estee Torok |
| | Ian G Goodfellow |
| | Cordelia Langford |
| | Dominic P Kwiatowski |
| NIHR | Gerry Tonkin-Hill |
| | Inigo Martincorena |
| | Roberto Amato |
| | Andrew RJ Lawson |
| | Moritz Gerstrung |
| | Ian Johnston |
| | David K Jackson |
| | Naomi Park |
| | Stefanie V Lensing |
| | Michael A Quail |
| | Sonia Gonçalves |
| | Cristina Ariani |
| | Michael Spencer Chapman |
| | William L Hamilton |
| | Luke W Meredith |
| | Grant Hall |
| | Aminu S Jahun |
| | Yasmin Chaudhry |
| | Myra Hosmillo |
| | Malte L Pinckert |
| | Iliana Georgana |
| | Anna Yakovleva |
| | Laura G Caller |
| | Sarah L Caddy |
| | Theresa Feltwell |
| | Fahad A Khokhar |
| | Charlotte J Houldcroft |
| | Martin D Curran |
| | Surendra Parmar |
| | Alex Alderton |
| | Rachel Nelson |
| | Ewan M Harrison |
| | John Sillitoe |
| | Stephen D Bentley |
| | Jeffrey C Barrett |
| | M Estee Torok |
| | Ian G Goodfellow |
| | Cordelia Langford |
| | Dominic P Kwiatowski |
| Wellcome | Gerry Tonkin-Hill |
| | Inigo Martincorena |
| | Roberto Amato |
| | Andrew RJ Lawson |
| | Moritz Gerstrung |
| | Ian Johnston |
| | David K Jackson |
| | Naomi Park |

Stefanie V Lensing
Michael A Quail
Sonia Gonçalves
Cristina Ariani
Michael Spencer Chapman
William L Hamilton
Luke W Meredith
Grant Hall
Aminu S Jahun
Yasmin Chaudhry
Myra Hosmillo
Malte L Pinckert
Iliana Georgana
Anna Yakovleva
Laura G Caller
Sarah L Caddy
Theresa Feltwell
Fahad A Khokhar
Charlotte J Houldcroft
Martin D Curran
Surendra Parmar
Alex Alderton
Rachel Nelson
Ewan M Harrison
John Sillitoe
Stephen D Bentley
Jeffrey C Barrett
M Estee Torok
Ian G Goodfellow
Cordelia Langford
Dominic P Kwiatowski

The funders had no role in study design, data collection and interpretation, or the decision to submit the work for publication.

## Author contributions

Gerry Tonkin-Hill, Data curation, Software, Formal analysis, Validation, Visualization, Methodology, Writing - original draft; Inigo Martincorena, Data curation, Software, Formal analysis, Supervision, Validation, Visualization, Methodology, Writing - original draft; Roberto Amato, Andrew RJ Lawson, Moritz Gerstung, Data curation, Software, Formal analysis, Methodology, Writing - review and editing; Ian Johnston, Data curation, Investigation; David K Jackson, Naomi Park, Stefanie V Lensing, Sónia Gonçalves, Cristina Ariani, Michael Spencer Chapman, Luke W Meredith, Grant Hall, Aminu S Jahun, Yasmin Chaudhry, Myra Hosmillo, Malte L Pinckert, Iliana Georgana, Anna Yakovleva, Laura G Caller, Sarah L Caddy, Theresa Feltwell, Fahad A Khokhar, Martin D Curran, Surendra Parmar, Alex Alderton, Rachel Nelson, John Sillitoe, Investigation; Michael A Quail, Investigation, Methodology; William L Hamilton, Charlotte J Houldcroft, Investigation, Writing - review and editing; The COVID-19 Genomics UK (COG-UK) Consortium, Wellcome Sanger Institute COVID-19 Surveillance Team, Data curation, Funding acquisition, Resources, Software, Project administration; Ewan M Harrison, Investigation, Project administration; Stephen D Bentley, M Estee Torok, Ian G Goodfellow, Cordelia Langford, Supervision, Writing - review and editing; Jeffrey C Barrett, Writing - review and editing; Dominic Kwiatkowski, Conceptualization, Supervision, Methodology, Writing - original draft, Project administration, Writing - review and editing

## Author ORCIDs

Gerry Tonkin-Hill https://orcid.org/0000-0003-4397-2224
David K Jackson http://orcid.org/0000-0002-8090-9462
Michael Spencer Chapman http://orcid.org/0000-0002-5320-8193
William L Hamilton http://orcid.org/0000-0002-3330-353X
Grant Hall http://orcid.org/0000-0003-3928-3979
Aminu S Jahun http://orcid.org/0000-0002-4585-1701
Myra Hosmillo http://orcid.org/0000-0002-3514-7681
Iliana Georgana http://orcid.org/0000-0002-8976-1177
Sarah L Caddy http://orcid.org/0000-0002-9790-7420

Charlotte J Houldcroft ⓘ http://orcid.org/0000-0002-1833-5285
M Estee Torok ⓘ http://orcid.org/0000-0001-9098-8590
Ian G Goodfellow ⓘ http://orcid.org/0000-0002-9483-510X

## Ethics

Human subjects: This study was done as part of surveillance for COVID-19 under the auspices of Section 251 of the National Health Service Act 2006. It therefore did not require individual patient consent or ethical approval. The COVID-19 Genomics UK (COG-UK) study protocol was approved by the Public Health England Research Ethics Governance Group.

## Decision letter and Author response
Decision letter https://doi.org/10.7554/eLife.66857.sa1
Author response https://doi.org/10.7554/eLife.66857.sa2

# Additional files

## Supplementary files
• Supplementary file 1. Sample IDs and metadata.
• Supplementary file 2. Shearwater within-host variant calls.
• Supplementary file 3. Sample specific sequence accession numbers.
• Supplementary file 4. Recurrent within-host mutations.
• Transparent reporting form

## Data availability
Sequencing data have been deposited in the ENA under the accession code ERP126512. All metadata has been provided in supplementary files.

The following datasets were generated:

| Author(s) | Year | Dataset title | Dataset URL | Database and Identifier |
|---|---|---|---|---|
| Tonkin-Hill G | 2021 | Patterns of within-host diversity in 1,181 SARS-CoV-2 samples sequenced to high depth in duplicate | https://www.ncbi.nlm. nih.gov/bioproject/ 694422 | NCBI BioProject, PRJEB42623 |
| Tonkin-Hill G | 2021 | Code and data to accompany the manuscript: "Patterns of within-host genetic diversity in SARS-CoV-2" | http://doi.org/10.5281/ zenodo.5115287 | Zenodo, 10.5281/ zenodo.5115287 |

The following previously published dataset was used:

| Author(s) | Year | Dataset title | Dataset URL | Database and Identifier |
|---|---|---|---|---|
| Tonkin-Hill G | 2020 | COVID-19 Genomics UK (COG-UK) consortium | https://www.ebi.ac.uk/ ena/browser/view/ PRJEB37886 | European Nucleotide Archive, PRJEB37886 |

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
