## [Decision Letter]

**Acceptance summary:**

Tonkin-Hill and colleagues present a large set of deep sequencing data from acute SARS-CoV-2 infections with each sample sequenced in duplicate. They use these data to characterize the within-host mutational patterns and diversity and relate them to SARS-CoV-2 diversity in consensus sequences sampled around the globe. It further allows understanding how this variation can or cannot be used to understand transmission dynamics and other applications in genomic epidemiology. The authors also provide extensive raw and processed data that can serve as a basis for further analysis of intra-host variation of SARS-CoV-2.

**Decision letter after peer review:**

Thank you for submitting your article "Patterns of within-host genetic diversity in SARS-CoV-2" for consideration by *eLife*. Your article has been reviewed by 3 peer reviewers, including Richard A Neher as Reviewing Editor and Reviewer #1 and the evaluation has been overseen by Sara Sawyer as the Senior Editor. The following individual involved in review of your submission has agreed to reveal their identity: Adam S Lauring (Reviewer #3).

Essential revisions:

All reviewers agreed that you present a very valuable dataset of impressive size and quality, that illuminates many important questions of within-host diversity. But the reviewers have made a number of suggestions to improve presentation and clarity.

1) Dependence of diversity measures on thresholds. We suggest adding a graph that shows the fraction of variable sites at first, second, and third position of codons above frequency x as a function of the threshold x. The threshold above which biological variation starts dominating over technical variation is usually clearly visible in such a graph. Furthermore, a discussion of the effect on thresholds on the conclusions (e.g. 0.72 variants per sample on average) and the expected number of false positives is necessary. The dependence of within-sample variation on Ct values should also be discussed.

2) Please add a detailed description of the sequencing methods and replication procedure.

3) The work would benefit from a better contextualization in the existing literature of RNA virus within-host variation and mutational processes. The authors have reinvented a fair bit of existing knowledge (several suggestions and pointers can be found in the specific comments below).

4) This work has the potential to serve as a reference resource of SARS-CoV-2 within-host variation and reuse of these data would be greatly facilitated by intermediate files, for example, tabular files for each sample listing the number of times each nucleotide is observed at each position of the genome.

5) Please consider the numerous suggestions to improve presentation and clarity.

*Reviewer #1 (Recommendations for the authors):*

The authors seem unaware of much of the literature on RNA intra-host sequencing and repeatedly reinvent the wheel. In fact, they often make reference to cancer, bacterial, or DNA virus genomics when placement in the literature of RNA virus genomics would be much more appropriate. A detailed discussion of mutational biases of SARS-CoV-2 and the potential RNA editing enzymes involved can be found here:

https://msphere.asm.org/content/5/3/e00408-20

In contrast, mutational mechanisms relevant in cancer are unlikely relevant here (nuclear dsDNA genome vs cytoplasmic RNA).

The quantification of biases and errors in estimates of within-host variation from replicate samples was for example developed pretty much along the same lines in

https://academic.oup.com/ve/article/3/2/vex030/4629376

http://www.sciencedirect.com/science/article/pii/S0168170216304221

https://academic.oup.com/ve/article/5/1/vey041/5304643

https://jvi.asm.org/content/90/15/6884

Within-host diversity is commonly used to estimate time since infection in HIV, both in acute and chronic infection.

https://link.springer.com/article/10.1186/1471-2105-11-532

https://academic.oup.com/cid/article/52/4/532/380068

http://journals.plos.org/ploscompbiol/article?id=10.1371/journal.pcbi.1005775

The observations with respect to within-host purifying selection are also very similar to what is observed in other RNA viruses (HIV, influenza virus, enteroviruses). See for example here:

https://academic.oup.com/ve/article/6/1/veaa010/5739536?login=true

Similarly, co/super-infection has been described in RNA viruses.

https://www.ncbi.nlm.nih.gov/pmc/articles/PMC136598/

https://academic.oup.com/ve/article/5/1/vez007/5479511

This is not to detract from the work the authors have done here, it just reads extremely odd when the authors describe a pattern in the data and then go on to explain how this is reminiscent of work done in bacterial genomics or cancer when it is in fact a common pattern in RNA viruses.

*Reviewer #3 (Recommendations for the authors):*

None of this precludes publication in *eLife*. This is a really well done study of an important topic.

1. Figure 1A and 1B shows only a random subset of 100 samples, which is far less than half of the dataset. This figure should be revised to include all of the samples (perhaps histograms of mutations per sample). This should also be broken down to show the number of within-host variants per sample by SNV, insertion/deletion, etc. Modifying this figure would showcase the data better and allow easier comparison to other studies.

2. With respect to variant calling. Again, the authors really should be commended for doing replicate samples. However, they should acknowledge that this limits, but does not remove, false positive variant calls (although it gets one to the level where truth can be hard to come by). From the manuscript it appears that they called variants in replicates down to a frequency of 0.5%. They acknowledge reduced sensitivity. But specificity is not perfect. It is worth looking at McCrone and Lauring JVI 2016, which also used DeepSNV and replicate samples. In this benchmarking study, the specificity at 0.5% frequency threshold was 99.99%. In a genome the size of influenza, this amounted to 3 false positives per sample (even with replicates). For a genome twice the size (SARS-CoV-2), one might expect up to 6 false positive variants. I don't expect the authors to reanalyze the data. But it is important that they discuss the potential that their summary statistics on number of single nucleotide variants per sample or per day could be inflated. Again, nothing wrong with that, but it helps to put this work in context with other data out there.

3. Overall, the variant calling is not clearly described – some in results, some in methods. But lots of details missing.

4. Others have shown (see Grubaugh Genome Biology, for example) that SNV within primer binding sites can cause aberrant frequency estimates. This can be an issue with large amplicon sets like the ARTIC protocol. It isn't clear how this was handled in this paper. Again, I don't think it changes the conclusions substantially.

5. I suspect that they are correct on the mutational/strand bias question and this analysis is fascinating to me. However, there are some important caveats. RT has its own mutational bias. While replicates should reduce the tendency for this to be a problem, it can impact this sort of analysis even in a limited way. The number of strands (plus vs. minus) could also play a role – but experimentally in terms of the degree to which subgenomic messages are sampled in their data. It might be helpful to see if the mutational bias is uniform across the genome or clustered in regions where subgenomic messages could be playing a role. Finally, there is evidence for asymmetric mutational bias in polymerases (see Pauly et al., *eLife* 2017, which also goes into RT error as well).

6. Is there a relationship between the diagnostic Ct value and the number of within-host variants per sample? This would be a helpful complement to the overdispersion analysis, which is slightly more abstracted. For example, it looks like over half of their samples have Ct > 24, which is where there is a significant up-tick in overdispersion. How will this affect their data on number variants identified and the number per day?

7. The analysis of paired samples is interesting, but the data is limited as the authors note. I am not convinced that there is a real increase in variants per sample over time. The difference in called variants across samples collected on the same day is quite large, in many cases larger than the increase in mutation abundance in the later timepoints. It is equally possible that there are limitations in variant calling sensitivity/specificity. What is the difference in Ct values for these paired samples?

8. The finding of recurrent mutations is nicely done. Several comments related to this:

a. It is interesting that D614G was seen multiple times. Do other recurrent within-host mutations identified in the paper reach high frequencies in GISAID data?

b. There are some sites that are prone to systematic errors, such as 11083 near a poly-U tract. How was this position handed in the analysis?

c. It would be good to have a table with each recurrent mutation and its counts.

d. Is it possible that any of these mutations are from cross-contamination? Are there fixed mutations at the same sites in other samples sequenced in the same batch? Or rather, how were replicates handled to minimize cross contamination?

e. I appreciate the difficulties in displaying multidimensional data. However, in Figure 5B, it is difficult to see the individual lines and pick out any patterns. There are too many overlapping lines. I would encourage trying other ways to display this figure, especially because it represents a central point of the paper.

---

## [Author Response]

Reviewer #1 (Recommendations for the authors):The authors seem unaware of much of the literature on RNA intra-host sequencing and repeatedly reinvent the wheel. In fact, they often make reference to cancer, bacterial, or DNA virus genomics when placement in the literature of RNA virus genomics would be much more appropriate. A detailed discussion of mutational biases of SARS-CoV-2 and the potential RNA editing enzymes involved can be found here:https://msphere.asm.org/content/5/3/e00408-20In contrast, mutational mechanisms relevant in cancer are unlikely relevant here (nuclear dsDNA genome vs cytoplasmic RNA).

We thank the reviewer for these recommendations. It is certainly true that the expertise of some of the authors is on bacterial evolution and cancer genomics, which influenced our choices of references. We are thus grateful for the reviewer’s comments and for the suggested references, which we have incorporated.

We now cite the article above on the possible role of APOBEC3 enzymes. It is important to note, that there is very limited knowledge on the mutation spectra caused by different APOBEC enzymes on RNA, as acknowledged by the article above. This article also cited the sequence context of the C>U mutations (using raw counts) as suggestive evidence of a role for APOBEC3. However, in our larger dataset we noticed that this context was much weaker when normalising the mutational spectrum by the trinucleotide sequence composition of the SARS-CoV-2 genome. Thus, while APOBEC and ADAR editing enzymes have been proposed to participate in the generation of mutations in SARS-CoV-2, a possibility that we also acknowledge, the evidence remains circumstantial as far as we can see. It remains possible that simple RNA damage (spontaneous cytosine deamination and guanine oxidation) rather than RNA editing can play a major role. Those two are likely damaging events in RNA molecules, and may be expected to cause the spectrum and strand asymmetries observed in SARS-CoV-2. They may also be more consistent with the modest sequence context of the C>U and G>T mutations observed. We mention both options as possible in our manuscript.

The quantification of biases and errors in estimates of within-host variation from replicate samples was for example developed pretty much along the same lines inhttps://academic.oup.com/ve/article/3/2/vex030/4629376http://www.sciencedirect.com/science/article/pii/S0168170216304221https://academic.oup.com/ve/article/5/1/vey041/5304643https://jvi.asm.org/content/90/15/6884

Thank you. We now cite these articles.

Within-host diversity is commonly used to estimate time since infection in HIV, both in acute and chronic infection.https://link.springer.com/article/10.1186/1471-2105-11-532https://academic.oup.com/cid/article/52/4/532/380068http://journals.plos.org/ploscompbiol/article?id=10.1371/journal.pcbi.1005775

We have added the line:

'estimates of within-host diversity have previously been used successfully to estimate the time since infection in HIV {Giorgi2010-wr,Kouyos2011-tw,Puller2017-qd}.'

The observations with respect to within-host purifying selection are also very similar to what is observed in other RNA viruses (HIV, influenza virus, enteroviruses). See for example here:https://academic.oup.com/ve/article/6/1/veaa010/5739536?login=true

We have added the line:

'This is similar to that observed in other RNA viruses such as Influenza {Xue2019-sn}.'

Similarly, co/super-infection has been described in RNA viruses.https://www.ncbi.nlm.nih.gov/pmc/articles/PMC136598/https://academic.oup.com/ve/article/5/1/vez007/5479511This is not to detract from the work the authors have done here, it just reads extremely odd when the authors describe a pattern in the data and then go on to explain how this is reminiscent of work done in bacterial genomics or cancer when it is in fact a common pattern in RNA viruses.

We thank the reviewer for these useful suggestions. We have added the line:

'Instances of co-infection are common in other RNA viruses including enterovirus and HIV {Dyrdak2019-xk,Liu2002-tq}.'

Reviewer #3 (Recommendations for the authors):None of this precludes publication in eLife. This is a really well done study of an important topic.

We thank the reviewer for the positive summary of our manuscripts and for the comments below.

1. Figure 1A and 1B shows only a random subset of 100 samples, which is far less than half of the dataset. This figure should be revised to include all of the samples (perhaps histograms of mutations per sample). This should also be broken down to show the number of within-host variants per sample by SNV, insertion/deletion, etc. Modifying this figure would showcase the data better and allow easier comparison to other studies.

We have modified Figure 1 to include boxplots indicating the distribution of the number of mutations by type for all samples. We have also modified figures 1C and D in line with reviewer 1’s comments.

2. With respect to variant calling. Again, the authors really should be commended for doing replicate samples. However, they should acknowledge that this limits, but does not remove, false positive variant calls (although it gets one to the level where truth can be hard to come by). From the manuscript it appears that they called variants in replicates down to a frequency of 0.5%. They acknowledge reduced sensitivity. But specificity is not perfect. It is worth looking at McCrone and Lauring JVI 2016, which also used DeepSNV and replicate samples. In this benchmarking study, the specificity at 0.5% frequency threshold was 99.99%. In a genome the size of influenza, this amounted to 3 false positives per sample (even with replicates). For a genome twice the size (SARS-CoV-2), one might expect up to 6 false positive variants. I don't expect the authors to reanalyze the data. But it is important that they discuss the potential that their summary statistics on number of single nucleotide variants per sample or per day could be inflated. Again, nothing wrong with that, but it helps to put this work in context with other data out there.

The 99.99% specificity reported in McCrone and Lauring, (2016) refers to the analysis of low input samples and higher specificity was observed with larger inputs. This is consistent with the increased discrepancy we observed in samples with high Ct values. To better acknowledge the potential limits of our approach we have added the following to the results:

'The use of replicates and a base-specific statistical error model for calling within-host diversity reduces the risk of erroneous calls at low allele frequencies. We noticed a slight increase in the number of within-host diversity calls for samples with high Ct values, which may be caused by a small number of errors or by the amplification of rare alleles and that could inflate within-host diversity estimates (Figure 1 —figure supplement 3) {McCrone2016-se}. However, the overall quality of the within-host mutation calls is supported by a number of biological signals. As described in the following sections, this includes the fact that the mutational spectrum of within-host mutations closely resembles that of consensus mutations and the observation of a clear signal of negative selection on within-host mutations, as demonstrated by dN/dS and by an enrichment of within-host mutations at third codon positions {Dyrdak2019-xk} (Figure 1 - figure supplement 4).'

3. Overall, the variant calling is not clearly described – some in results, some in methods. But lots of details missing.

We have now included the R scripts used to perform the variant calling under the

‘shearwater_variant_calling_scripts’ sub folder along with an accompanying readme file in the GitHub repository. We have also amended the methods to include a description of the alignment method used:

'Alignment was performed using the ARTIC Illumina nextflow pipeline available from {https://github.com/connor-lab/ncov2019-artic-nf}. The pipeline trims primer and adapter sequences prior to alignment to the reference genome using bwa (MN908947.3) {Li2009-fe}. Variant calling was performed using ShearwaterML and bam2R was used to count the number of reads supporting each base at each site. Both these methods are available as part of the deepSNV R package {Gerstung2014-av,Martincorena2015-ef}. All scripts used to perform the variant calling are available in the accompanying GitHub repository.'

4. Others have shown (see Grubaugh Genome Biology, for example) that SNV within primer binding sites can cause aberrant frequency estimates. This can be an issue with large amplicon sets like the ARTIC protocol. It isn't clear how this was handled in this paper. Again, I don't think it changes the conclusions substantially.

To prevent this problem primer regions were trimmed from reads prior to alignment and the calling of variants. The ARTIC protocol uses overlapping amplicon sets which allows for trimming while maintaining coverage across the full genome. We have added the following line to the methods to clarify this:

'The pipeline trims primer and adapter sequences prior to alignment to the reference genome using bwa (MN908947.3) {Li2009-fe}.'

5. I suspect that they are correct on the mutational/strand bias question and this analysis is fascinating to me. However, there are some important caveats. RT has its own mutational bias. While replicates should reduce the tendency for this to be a problem, it can impact this sort of analysis even in a limited way. The number of strands (plus vs. minus) could also play a role – but experimentally in terms of the degree to which subgenomic messages are sampled in their data. It might be helpful to see if the mutational bias is uniform across the genome or clustered in regions where subgenomic messages could be playing a role. Finally, there is evidence for asymmetric mutational bias in polymerases (see Pauly et al., eLife 2017, which also goes into RT error as well).

This is an interesting question. The reviewer rightly mentions a number of possible caveats, including RT and PCR biases, the error spectrum of the reverse transcriptase or the DNA polymerase used for PCR, and the possibility that subgenomic messages are sampled in the data. While all of these factors could potentially influence the within-host spectrum, particularly if there are residual errors among the mutation calls, the strongest argument against these factors having a major influence is that the within-host spectrum closely resembles that of consensus or fixed inter-host differences (see Figure 2B and D). That is, the asymmetries described in our manuscript (and in previous studies) are equally strong when restricting the analysis to very high-confidence consensus mutations (e.g. supported by ~100% of reads and restricted to low Ct samples).

6. Is there a relationship between the diagnostic Ct value and the number of within-host variants per sample? This would be a helpful complement to the overdispersion analysis, which is slightly more abstracted. For example, it looks like over half of their samples have Ct > 24, which is where there is a significant up-tick in overdispersion. How will this affect their data on number variants identified and the number per day?

We have added an additional plot indicating the number of within-host variants by Ct value. This indicates that there is an uptick in the number of calls for samples with high Ct. We have also described this in the text as shown in the reply to point 2 above.

7. The analysis of paired samples is interesting, but the data is limited as the authors note. I am not convinced that there is a real increase in variants per sample over time. The difference in called variants across samples collected on the same day is quite large, in many cases larger than the increase in mutation abundance in the later timepoints. It is equally possible that there are limitations in variant calling sensitivity/specificity. What is the difference in Ct values for these paired samples?

We have amended Figure 3 to indicate the maximum Ct value for each point in the plot of paired samples. While it is not possible to eliminate the possibility that the weak signal is driven by noise, the trend is still visible in low Ct pairs. Similar trends are also observed in other RNA viruses including HIV. To better reflect that false positives could be contributing to the observed variation we have also added an additional caveat to the text:

“To put this finding in context, there was considerable variation in the observed number of within-host variants among samples from the same individual, even if taken on the same day, possibly as a result of samples with high Ct values as well as the bottleneck caused by the different sampling methods which included sputum, swabs and bronchoalveolar lavage (Figure 3 —figure supplement 1).”

8. The finding of recurrent mutations is nicely done. Several comments related to this:a. It is interesting that D614G was seen multiple times. Do other recurrent within-host mutations identified in the paper reach high frequencies in GISAID data?

We have added the frequency of non-reference mutations in the COG-UK dataset to a new supplementary table describing the recurrent mutations and amended the results to read:

“Still, this analysis does not rule out the possibility that a minority of recurrent within-host variants are the result of convergent positive selection. Indeed, a number of recurrent within-host variants are also observed to frequently occur at the consensus level in the broader COG-UK dataset (Supplementary Table 4). A plausible example of positive selection is the spike glycoprotein mutation D614G, which has rapidly increased in frequency throughout the world: within-host variation at this position appears in 27 of our samples with the D614G mutation typically found as the consensus variant.”

b. There are some sites that are prone to systematic errors, such as 11083 near a poly-U tract. How was this position handed in the analysis?

As it is not always possible to distinguish systematic errors from highly recurrent mutations we took a conservative approach and allowed the same mutation to contribute at most 5 times to the within-host mutational spectrum. This was chosen to avoid a single or a few hotspots or recurrent artefacts from distorting the spectrum of recurrent mutations. In addition, recurrent mutations seen in more than 2% of samples were filtered out in Figure 5B and in the transmission analysis (Figure 6B/C). The high number of shared within-host variants across the consensus phylogeny remained indicating that even after filtering out these sites caution is needed when using within-host to inform transmission estimates. This is described in the caption of Figure 5 and in the methods section for the mutational spectrum analysis. We have also added the following line to the methods section of the transmission analysis:

“Within-host variants observed in more than 2\% of samples were excluded from the histograms in 6B and C.”

c. It would be good to have a table with each recurrent mutation and its counts.

We have now added an additional supplementary table with these counts.

d. Is it possible that any of these mutations are from cross-contamination? Are there fixed mutations at the same sites in other samples sequenced in the same batch? Or rather, how were replicates handled to minimize cross contamination?

It is impossible to rule out that some of the shared mutations are due to contamination. However the association between the consensus phylogeny and instances of co-infection/contamination indicate any contamination would have to be structured and thus is likely to have occurred within the hospital. To address this we state that:

“While it is not possible to rule out cross-contamination between samples, this contamination would have to be structured to explain the correlations between the inferred co-infections and the consensus phylogeny. Batch effects were carefully controlled for by Lythgoe et al. and as we identified a similar signal using an independent dataset with samples sequenced in replicate, structured superinfection provides a better explanation.”

e. I appreciate the difficulties in displaying multidimensional data. However, in Figure 5B, it is difficult to see the individual lines and pick out any patterns. There are too many overlapping lines. I would encourage trying other ways to display this figure, especially because it represents a central point of the paper.

We have amended Figure 5 to try and better distinguish the underlying patterns. By separating the links based on the number of shared variants rather than using colour we feel that it is easier to interpret the figure. The large number of samples sharing a single within-host variant still leads to many overlapping lines but we feel that this represents the complexity of the dataset.